# Reversing Cardiac Hypertrophy at the Source Using a Cardiac Targeting Peptide Linked to miRNA106a: Targeting Genes That Cause Cardiac Hypertrophy

**DOI:** 10.3390/ph15070871

**Published:** 2022-07-15

**Authors:** G. Ian Gallicano, Jiayu Fu, Samiksha Mahapatra, Michael V. R. Sharma, Conor Dillon, Claire Deng, Maliha Zahid

**Affiliations:** 1Department of Biochemistry and Molecular Biology, Georgetown University Medical Center, 3900 Reservoir Rd, Washington, DC 20057, USA; jf1465@georgetown.edu (J.F.); smahapatra@health.ucsd.edu (S.M.); cyd7@georgetown.edu (C.D.); 2Celprogen, Torrance, CA 90503, USA; jaysharma@celprogen.com; 3Bishop O-Connell High School, 6600 Little Falls Rd, Arlington, VA 22213, USA; cindyodillon@gmail.com; 4Department of Cardiovascular Medicine, Mayo Clinic, Rochester, MN 55905, USA; maz7@pitt.edu; 5Chief Scientific Officer, Vivasc Therapeutics, Inc., Pittsburgh, PA 15203, USA

**Keywords:** heart failure, miRNA therapy, hypertrophy, cardiac targeting peptide

## Abstract

Causes and treatments for heart failure (HF) have been investigated for over a century culminating in data that have led to numerous pharmacological and surgical therapies. Unfortunately, to date, even with the most current treatments, HF remains a progressive disease with no therapies targeting the cardiomyocytes directly. Technological advances within the past two to three years have brought about new paradigms for treating many diseases that previously had been extremely difficult to resolve. One of these new paradigms has been a shift from pharmacological agents to antisense technology (e.g., microRNAs) to target the molecular underpinnings of pathological processes leading to disease onset. Although this paradigm shift may have been postulated over a decade ago, only within the past few years has it become feasible. Here, we show that miRNA106a targets genes that, when misregulated, have been shown to cause hypertrophy and eventual HF. The addition of miRNA106a suppresses misexpressed HF genes and reverses hypertrophy. Most importantly, using a cardiac targeting peptide reversibly linked to miRNA106a, we show delivery is specific to cardiomyocytes.

## 1. Introduction

Etiologies of heart failure (HF) have been investigated for over a century culminating in data that have led to numerous pharmacological and surgical therapies. Unfortunately, to date, even with the most current treatments, the progressive disease remains associated with an annual hospitalization rate of ~1 million and a mortality rate of ~400,000 people (in the U.S. [1]), and for those that live with the disease, only marginally improve quality of life. Technological advances within the past two to three years have brought about new paradigms for treating many diseases that previously had been extremely difficult to resolve. One of these new paradigms has been a shift from pharmacological agents to antisense technology (e.g., microRNAs) that molecularly suppress disease onset. Although this paradigm shift may have been postulated over a decade ago [2], only within the past few years has it become feasible. A prime example illustrating the clinical implementation of this paradigm is the recent (2017) antisense agent IONIS-HTTRx (Ionis Inc., Carlsbad, CA, USA) for treating Huntington’s disease HTT [3]. This therapeutic approach is showing great promise not only with respect to safety but also in clearing away all mutant forms of HTT protein in phase 1 and 2 clinical trials. Antisense technologies are now becoming feasible for treating cancers [4], muscular dystrophy [3], some viral infections [5,6], and, as we propose here, HF.

MiRNAs are small, non-coding, RNA molecules (containing about 22 nucleotides) found in plants, animals, and some viruses. Their main function is in mRNA silencing, with post-transcriptional regulation of gene expression controlled by targeting the 3′UTR of an mRNA through a seed sequence consisting of 7–8 bases [7,8,9,10,11]. MiRNA/siRNA use in clinical settings has been beset by a few challenges. At the time when miRNAs were proposed for potential clinical applications in HF [2], one distinct challenge was a method of delivery to the heart. Research over the past decade brought about a multitude of methodologies all with varying levels of success and failure. One prevailing technology has been the discovery of a cardiac targeting peptide (CTP) that can be reversibly linked (e.g., via enolases, thiol groups) to cargo (e.g., miRNAs) for delivery specifically to cardiomyocytes over all other organs [12,13,14].

The second set of challenges has been to identify the misexpressed genes in HF to be targeted and at the same time, identify which miRNAs would be best suited for targeting those misexpressed genes. Again, a decade ago, the knowledge base was not advanced enough to confront this challenge; however, as miRNA/mRNA targeting profiles have been identified along with the generation of novel miRNA/mRNA prediction algorithms, it is now possible to resolve molecularly the problem(s) brought on by genes that cause HF.

The overall published mechanism(s) of HF is complex, with multiple different pathways that are not fully understood. Three genes/proteins, in particular, CaMKIIδ, HDAC4, and GRK2 have been shown to act as central hubs or be directly involved in the induction of cardiac hypertrophy when misexpressed [15,16,17]. Other genes/proteins may also be involved with HF including PKA, STAT3, FOG2, and the family with sequence similarity 210 member A (FAM210A) [18,19,20,21,22,23]. Interestingly, the literature does highly suggest that most of the genes in HF pathways are intimately involved with each other either upstream or downstream with CaMKIIδ (both splice variants δ_**B**_ and δ_**C**_ [24]) serving as one of the core/hub proteins in HF [17,25,26,27,28,29].

With many of these ‘challenges’ now being perceived more as advancements, we believe the paradigm can now be shifted from treating HF using pharmacological agents to a molecular approach using miRNAs to restore cardiac physiological and structural identity. Based on our previous investigations showing that miRNAs 17, 20a, 93, and 106a regulated cardiac differentiation [20], and that they targeted (either directly or indirectly) genes that were also found in adult HF, we tested if these miRNAs could reverse an in vitro model of HF. In [20], ablation of miRNA106a was found to be the most deleterious to cardiac differentiation; however, because miRNA106a was able to dose-dependently rescue cardiac identity, we chose this miRNA to rescue hypertrophic cardiomyocytes.

As a result of our current research, the hypothesis to be tested is that it is now possible, at the molecular level, to not only inhibit the detrimental physiological and structural pathways of the failing heart but also induce an environment to reestablish the correct patterning of cardiomyocytes. Here, we show strong evidence that when miRNAs are chemically linked (reversibly) to CTP, they are not only delivered specifically to cardiomyocytes, but can also rescue/revert hypertrophic human cardiomyocytes (HCMs) without any noticeable cytotoxicity.

## 2. Materials and Methods

HCMs were purchased from Celprogen (Torrence, CA, USA). They are isolated from a punch biopsy from a donated human heart. HEK293 cells were purchased from ATCC. HUVECs were supplied by the Tissue Culture & Bio-banking Shared Resource (TCBSR) at Lombardi Comprehensive Cancer Center (Georgetown University, Washington, DC, USA). The TCBSR purchases all HUVECS from ATCC. Adult human cardiac fibroblasts were purchased from Sigma Aldrich (St. Louis, MO, USA; Cat #306-05A).

Angiotensin 2 (Ang2) and phenylephrine (PE) experiments. Cell was grown to 50% confluency and then treated with 10 nM Ang2 (Preprotech, Inc., east Windsor, NJ, USA) and 100 nM PE (Sigma Aldrich, Inc., St. Louis, MO, USA) for 24, 48, 72, 96, or 144 h Ang2 and PE bind to the Ang2 receptor and β-adrenergic receptors, respectively. Constitutive activation of these receptors leads to hypertrophy. Transfection of miRNAs, HCMs HEK293 cells was accomplished using *Trans*IT-X2**^®^** Dynamic Delivery System (cat# MIR6000, Madison, WI, USA) following the kit protocol. Stock miRNAs are prepared in RNAse-free ddH20 at 20 uM and transfected at 200 nM.

### 2.1. Western Blot Analyses

Western blots were performed using precast 4–20% Tris-HCL polyacrylamide gels and run on the BioRad mini gel system (Hercules, CA, USA). 5 × 10^5^ cells for all experiments were scraped and transferred to 200 μL Laemmle Sample buffer and loaded at 10 μL/well. After SDS-PAGE, proteins were transferred to PVDF membranes and blocked in a solution of 2% BSA and 5% milk in PBS-T. Antibodies [desmin (Elabscience, Inc., Houston, TX, USA), CamKIIδ (Abnova, Inc., Walnut, CA, USA), HDAc4, cTnnT, cardiac actin, ActB, GRK2, ubiquitin (Santa Cruz, Inc., Santa Cruz, CA, USA)] were all used at a 1:1000 dilution. Bands were visualized using an ImageQuant LAS4000 (General Electric, Inc., Philadelphia, PA, USA) analyzer. Similar exposure times were used for all antibodies and their ActB or desmin controls) to ensure appropriate normalization. Quantification of bands was performed using Adobe Photoshop 2021 to calculate the pixel intensity of each band. Pixel intensities for are expressed as ratios to generate a normalized value for each protein analyzed. All PVDF membranes were also normalized using Ponceau S staining as shown in Figure 1I. Mitofusin-2, a mitochondrial membrane, structural maintenance protein was analyzed using Mnf2 monoclonal antibody from Santa Cruz, Inc.

### 2.2. CTPmiRNA106a

The generation of the CTPmiRNA106a was based on previous publications [12,13]. Briefly, Cy5.5 labeled cardiac targeting peptide (CTP) was synthesized on a Liberty CEM microwave synthesizer using Fluorenylmethyloxycarbonyl (FMOC) chemistry. The C-terminal carboxyl group of CTP was conjugated with Cy5.5-amine (Lumiprobe Corporation, Cockeyeville, MD, USA) using DIPCDI/DMF activation. Final cleavage of CTP-Cy5.5 with Trifluoroacetic acid: Triisopropylsilane: H2O (TFA:TIPS:H2O) (90:25:25) was followed by precipitation in EtO2. MALDI-Tof analysis of the purified conjugate was performed on an Applied Biosystems Voyager workstation using α-Cyano-4-hydroxycinnamic acid (CHCA) matrix allowing for confirmation of the expected mass and identity of the final CTP-Cy5.5 conjugate.

Conjugation of miRNA106a to CTP was performed by synthesizing 10 mg of miRNA106a (General Biosystems, Inc., Durham, NC, USA) with a thiol modified 5′ end was linked to CTP. After cleavage of the fully side-chain protected CTP peptide from the 2-chlorotrityl resin under mildly acidic conditions, it was followed by the N-terminus being thiolated with 2-iminothiolane (Traut’s reagent) in 0.1 M Triethylammonium bicarbonate (TEAbc) at pH 8.5. The resulting thiolated CTP peptide was then directly purified by preparative C-18 RP-HPLC on a Waters Delta Prep 4000 chromatography system using standard Acetonitrile/0.1%TFA gradient conditions followed by lyophilization. Thiol-modified miRNA oligomers were reduced to their free thiol form using DL-Dithiothreitol (DTT) in 0.1 M TEAbc at pH 8.5 and then reacted with dithio-bis-maleimidoethane (DTME) in 300 mM NaOAc/acetonitrile at pH 5.2. Purification of the siRNA-DTME intermediate was then followed by reaction with the purified CTP peptide in 300 mM NaOAc/acetonitrile at pH 5.2 with gentle mixing at room temperature. Analytical C-18 RP-HPLC purification of the resulting miRNA106a-DTME-CTP-Cy5.5 conjugates using trimethylamine acetate (TEAA)/Acetonitrile gradients on a Waters Alliance chromatography system was followed by lyophilization and then re-lyophilization from nuclease-free water. MALDI-Tof analysis of the purified conjugate on an Applied Biosystems Voyager workstation using 3-hydroxypicolinic acid (3-HPA) matrix in ammonium citrate allowed for confirmation of the expected mass and identity of the final miRNA-CTP conjugate.

### 2.3. FACs and Immunofluorescence

HCMs or HEK293 cells were incubated in Cy5.5-CTP-miRNA106a for 30 min and then sorted at the Cy5.5 wavelength. Samples for flow cytometry were fixed in 4% formaldehyde followed by permeabilization with 0.1% Triton-X-100. Flow cytometry data were acquired and analyzed by members of the Georgetown University Flow Cytometry Core Facility using a BD LSRFortessa Cell Analyzer, the Cat. No. is 647177.

For immunofluorescence, cells were grown on 0.1% fibronectin coverslips, fixed in 4% formaldehyde followed by permeabilization with 0.1% Triton-X-100 and then blocked in PBS+2% bovine serum albumen+1% Tween 20 for 1 h Desmin (Elabscience, Inc. Cat# E-AB-31185), HDAc4 (Santa Cruz, Inc. sc-365367), vimentin (Abcam, Inc., Cambridge, UK; ab8978) were added to block at 1:100 dilution and added to cells for 1 h or overnight, followed by 1 h wash in block and 1 h in Alexa Fluor (Molecular Probes, Inc., Eugene, OR, USA). Dapi was added for 30 min to amplify nuclear staining. Slides were viewed using an Olympus Fluoview 500 four channel Laser Scanning Microscope (Olympus America Inc, Melville, NY, USA) using 1.4 numerical aperture. Images were acquired and analyzed using the accompanying Fluoview software (version 4.4., Olympus America, Melville, NY, USA).

### 2.4. Immunoprecipitation Assay

To identify GRK2 ubiquitination, we used the protocol from [30], which encompassed generating lysates from an equal number of cells and subjecting them to mouse anti-ubiquitin for 1 h at 4 °C. antibodies were isolated using magnetic, G-protein-coated beads. After SDS-PAGE and transfer to PVDF, rabbit anti-GRK2 was used as probe. Using a secondary generated in a different animal as the primary minimizes cross-reactivity. Membranes were analyzed using the ImageQuant LAS4000 machine.

### 2.5. Luciferase Assays

Luciferase assays were conducted in HEK293 cells by transient transfections of pRL-CMV-HDAc4, CamKIIδ, or GRK2 3′UTRs, followed 24 h later by transfection of 200 nM miRNA106a or 5 ug/mL CTPmiRNA106a. Cells were harvested according to the specifications of the Dual Luciferase Reporter Assay Kit (Promega, Madison, WI, USA) and 20 uL of the lysate was transferred to each well of a 96-well plate for analysis. The luciferase reactions were conducted using the Wallac VICTOR^2^ 96-well plate reader at the Lombardi Cancer Center Shared Resource. All luciferase data are presented as a normalized ratio of luciferase/renilla.

### 2.6. Mitochondria Analyses

JC-1 dye (Fisher Scientific, Inc., Waltham, MA, USA) was used to probe mitochondrial membrane potential. As described in [31], cells were incubated in 5 ug/mL of JC-1 dye for 20 min at 37 °C. Using a fluorescence microscope, JC-1 monomers or aggregates were detected as green, unhealthy monomer puncta, or red, healthy puncta. Red and green ratios were obtained using the pixel intensity tool within Photoshop. Microscope settings were normalized to untreated cells. The entire cytoplasm was outlined resulting in an intensity level from 0 (black) to 255 (highest pixel intensity). All ratios were then normalized to untreated (ratio of 1). All experiments were performed at least six times.

### 2.7. qRT-PCR Analyses

After transfection of miRNA106a into HCMs or HEK293 cells via the CTPmiRNA106a or traditional transfection using *Trans*IT-X2 transfection reagent, RNA was isolated from three different batches (biological replicates; *n* = 3) using RNEasy isolation kit from Qiagen (Cat #217004). Using the miScript Sybr green kit (Qiagen Inc., Germantown, MD, USA), which attaches a hairpin loop RNA to the 3′ end of miRNAs. MiRNA was specifically amplified using the miRNA106a sequence GUGCUUACAGUGCAGGUAG and the universal primer in the kit. Amplification and change in thresholds (dcT) were obtained using a Bio-Rad CFX Connect real Time PCR system. The ddCT was then calculated and graphed.

### 2.8. Quantification and Statistics

Pixel intensity of each Western blot band was measured using the Rectangular Marquee and Histogram tool in Adobe Photoshop 2022. Results are obtained as pixel intensity (255 maximum intensity) of experimental protein/control protein ratio to get a normalized protein expression for each band, followed by a comparison of each band ratio to time 0 (untreated). All experiments were done at least six times (*n*= 6) or more. Significance or insignificance were obtained using paired *t*-tests comparing a treated set of HCMs to untreated. To ensure reproducibility and validity of the data, we followed statistical analyses suggested by [32,33,34,35,36]. Pixel intensity was obtained for immunofluorescence analyses similar to Western blot analyses except for the area of a cell or nucleus, which was outlined using the Lasso tool in Photoshop.

## 3. Results

Human cardiomyocytes (HCMs) were subjected to 200 nM phenylephrine (PE) and 10 nM angiotensin 2 (Ang2) to induce hypertrophy (initial stages of HF) [37]. Using *Trans*IT-X2 transfection reagent, 200 nM of each miRNA, or in combination, was introduced into PE/Ang2-treated HCMs, followed by analyses of hypertrophic morphology and HF gene/protein expression. A total of 200 nM concentration was chosen based on its efficacy from numerous dosage comparisons found in previous publications [8,19]. The data in Figure 1A–E provide evidence that these miRNAs can significantly rescue/revert hypertrophic HCMs back to their normal phenotypic dimensions even in the presence of PE/Ang 2 treatment.

The next steps were to (1) identify if genes involved with hypertrophy and HF were targets for the four miRNAs and (2) ascertain the downstream signaling events that would be affected by miRNA targeting. CaMKIIδ, HDAC4, and GRK2 have been shown to act as central hubs or be directly involved in the induction of cardiac hypertrophy when misexpressed (Figure 1F) [15,16,17,25,26,27,28,29,38,39,40]. Using the microRNA database, www.miRDB.org (accessed 13 July 2022) to identify in silico miRNA/mRNA targeting, we found CaMKIIδ and HDAC4 both were direct targets for miRNA106a (and miR17, 20a, and 93). Both genes have two miRNA106a target sites (Figure 1G,H). Western blot analyses showed that each of these four miRNAs suppressed CaMKIIδ and HDAC4 overexpression when induced by PE/Ang2 suggesting these genes are targets for miRNAs 17, 20a, 93, and 106a (Figure 1I,J). Cardiac actin (no 3′UTR target) expression was measured as a hypertrophy control. As cells increase in size they require more cardiac actin, which is reduced as they revert to normal size after treatment with each miRNA (Figure 1K).

We previously verified that during cardiac development, STAT3 was a direct target of miRNAs 17, 20a, 93, and 106a when analyzed using Western blots and 3′UTR luciferase assays [9,20,41]. Using similar techniques to verify that HDAC4 and CaMKIIδ were miRNA targets, full-length 3′UTRs for both genes were fused to firefly/renilla luciferase constructs. Analyses provided conclusive evidence that at least one of the miRNA family, miRNA106a, targets the 3′UTRs of both genes (Figure 2A,B). The 3′UTR for GRK2 was used as a control because it does not have a target sequence for miRNA106a. It showed no change in luciferase expression.

*Identifying CTP for delivery of miRNAs to HCMs***:** CTP was linked through a disulfide bond with one of the miRNAs, miRNA106a, for delivery to hypertrophic HCMs. Figure 3 is a summary of multiple experiments illustrating (**1**) CTP targets HCMs preferentially compared with the HEK293 cells (a high transfection efficiency cell line) and other cardiac cell types (e.g., cardiac fibroblasts and endothelial cells) and (2) CTP can deliver miRNA106a specifically to HCMs. Figure 3A–D shows the fluorescent Cy5.5 marker linked to CTP targeting to HCMs and not HEK293 cells 1 h after incubation. We chose HEK293 cells specifically as a control because they are considered easy to transfect; however, very little CTP is observable in HEK293 cells (Figure 3A–D). Molecular analyses using qrtPCR demonstrated that 5 ug/mL CTPmiRNA106a (equivalent to 200 nM) delivered ~2.5X more miRNA106a to cardiomyocytes than transfecting 200 nM miRNA106a using *Trans*IT-X2 transfection reagent (Figure 3E). The analyses for Figure 3E were done 24 h after transfection with either CTPmiR106a or transfection using a traditional transfection reagent. A third approach using FACS analyses confirmed the specificity of delivery of CTPmiRNA106a to HCMs (Figure 3F,G). Virtually no increase in miRNA106a was detected in non-cardiac HEK293 cells treated with CTP-miRNA106a. Three basic types of cells are found in the heart; cardiomyocytes, cardiac fibroblasts, and endothelial cells. Testing CTPmiRNA106a on these cell types, we found only a small signal overlap detected within endothelial cells was detectable (Figure 3H).

Next, the luciferase assay was repeated, but this time HCMs were subjected to CTP-miRNA106a, without using the *Trans*IT-X2 transfection reagent. Figure 2C,D provide strong evidence that the miRNA106a that was delivered by the CTP targeted both CaMKIIδ and HDAC4 3′UTRs, resulting in a significant decrease in luciferase expression. This result indicates that miRNA106a is physically free and bioactive. It must be processed correctly in the RNA inhibitory silencing complex (RISC) in order for miRNA to directly interact with its target mRNA and suppress the translation of luciferase. Second, we then identified that the miRNA106a delivered by CTP could suppress the translation of CaMKIIδ and HDAc4. Three different experiments (last three lanes of each Western blot) show CTP-miRNA106a clearly suppresses the translation of each protein (Figure 4). Downstream targets (and markers for hypertrophy) of CaMKIIδ and HDAC4 signaling, cardiac troponin (cTnnT), BNP, and GRK2 [19,28] were also analyzed to further identify the mechanism of action of CTP-miRNA106a in reverting the hypertrophic phenotype in HCMs. Western Analyses showed both BNP and cTnnT rising in response to PE and Ang2; however, both subsequently returned to baseline (untreated) levels in response to CTPmiRNA106a (Figure 4). BNP and cTNNT are not targets for miRNA106a.

HDAC4 localization was also rescued using CTPmiRNA106a. HDAC4 is nuclear in untreated HCMs; however, CaMKIIδ phosphorylation promotes its movement into the cytoplasm [16]. Figure 4C–F shows that transfection of miRNA106a using *Trans*IT-X2 or CTPmiRNA106a results in HDAC4 re-entry into the nucleus.

Finally, we analyzed the effects of CTP-miRNA106a on GRK2 signaling. GRK2 has been identified to cause HF when misexpressed. Figure 4G,H shows two Western blots where PE/Ang2 causes increases in GRK2 expression, followed by its return to baseline levels after transfection of each miRNA or culture in CTPmiRA106a. GRK2 is not a target of miRNAs 17, 20a, 93, or 106a; however, it was recently shown that GRK2 phosphorylation by CaMKIIδ prevents ubiquitination resulting in excess GRK2 [30]. To identify if ubiquitination levels of GRK2 were involved with these changes, we performed an immunoprecipitation assay similar to [30]. As predicted, PE/ANG2 treatment (which increased CaMKIIδ expression) led to a marked decrease in ubiquitin staining by 72 h of PE/Ang2 treatment; however, the addition of CTPmiRNA106a over time led to increased levels of ubiquitinated GRK2 (Figure 4).

We found one report suggesting that miRNA106a may cause cardiac hypertrophy by targeting and suppressing mitofusin 2 (Mfn2), a mitochondrial membrane protein involved in maintaining mitochondrial structure [31]. An interesting experiment in [31] was their analysis of mitochondrial health using the mitochondrial membrane potential (MMP) probe JC-1 staining. Healthy MMP is observed as JC-1 forms J-aggregates that fluoresce red (~594 nm) while unhealthy MMP is recognized as green (~488 nm) JC-1 monomers. We found no unhealthy (green MMPs) at any of the concentrations of Ang/PE, miRNA106a, or CTPmiRNA106a at the concentrations or timepoints we utilized throughout our experiments in Figure 1, Figure 2, Figure 3 and Figure 4 (Appendix A). However, we did begin to see green MMPs if we used 10X the amount of Ang2/PE or 10X CTPmiRNA106a for at least 72 h Western blots were used to identify Mfn2 as a target for 200 nM miRNA106a and 5 ug/mL CTPmiR106a. out of three dosages, significant (*p*
< 0.05) decreases in Mfn2 protein expression were only observed after incubation in 10× the concentration we normally use for experiments in Figure 1, Figure 2, Figure 3 and Figure 4 (Appendix A).

## 4. Discussion

Here, we find that four miRNAs, miR17, 20a, 93, and 106a are predicted to target genes known to cause HCM hypertrophy and HF (when misregulated) resulting in morphological and molecular reversion to basal (untreated) levels when transfected into hypertrophic HCMs. We found two primary results from the data; (1) that these miRNAs do target and suppress CaMKIIδ and HDAc4 and (2) that one of the miRNAs (106a) when reversibly linked to a CTP can be delivered to cardiomyocytes resulting in targeting and suppression of hypertrophic levels of CaMKIIδ and HDAc4.

Previously we showed the importance of miRNAs 17, 20a, 93, and 106a in cardiac development [9,10,20,41,42]. Most recently, we used CRISPR/CAS9 gene editing technology to knock-out each of the four miRNAs in embryonic stem cells [20]. Upon differentiation, we noticed that each miRNA knock-out resulted in changes in the expression of genes involved in heart development. Interestingly, a subset of those genes was also found to be associated with HF. For example, upon knocking out miRNA106a, we noticed significant changes in gene expression of brain natriuretic peptide (BNP), ventricular myosin heavy chain, and the gap junction protein Connexin 43 (CXN43). Importantly, when miRNA106a was added back to mir106a-/- ESCs, the expression of each gene returned to normal and cardiac differentiation was rescued.

For decades, numerous drugs have been developed and prescribed to ameliorate the end results of NH-induced HF. These drugs are usually classified with their corresponding target. For example, angiotensin receptor blockers antagonize angiotensin 2 type 1 receptors, while angiotensin-converting enzyme inhibitors attack the angiotensin converting enzyme and mineralocorticoid receptor antagonists inhibit the action of mineralocorticoids. Vessel tone/blood pressure/volume is affected by drugs including diuretics and vasodilators, while positive inotropes such as dobutamine are used to improve ventricular contraction. Surgical correction, including implanting devices, has also been an option along with pharmacological intervention in specific cases. However, even with the current ‘best treatments’, the underlying cause(s) of HF in these patients is not resolved and the end result is usually decreased quality of life or a shortened life span [43,44,45,46]. Current conventional drug therapy approaches do not resolve the underlying processes of HF. On the contrary, targeting genes that cause HF would enable hypertrophic HCMs to reestablish normal cardiac identity. The technology now exists for using RNA interference as a paradigm shift for resolving or even rescuing cardiac tissue undergoing HF.

Interestingly, [31] presented data suggesting miRNA106a induced hypertrophy by targeting and down-regulating Mnf2; however, we were not able to duplicate those results using our concentrations or dosages. Unfortunately, we could not find transfection concentrations or specified culture times for miRNA106a mimics or inhibitors used in [31] to duplicate their work. The dosages that we used for our experiments did not show significant loss of Mnf2 expression by Western blot or any mitochondrial membrane detriment as measured using the same JC-1 technique in [31].

We also noticed that the in vivo data in [31] showed a correlation between inhibition of miRNA106a and reversal of cardiac hypertrophy; however, their data were based on systemic injections into mice using a cholesterol moiety-linked miRNA106a inhibitor. There were no data showing that the miRNA106a-inhibitor was delivered to the heart. All results could have been due to inhibiting miRNA106a in other organs such as in endothelial cells where vasodilator-stimulated phosphoprotein (VASP) is a predicted target of miRNA106a (miRDB.org-Seed Location base 335-346 3′UTR). Mis-regulation of VASP in endothelial cells [47] could alter vessel tone resulting in changes in cardiac physiology and morphology as reported in [31]. Future experiments using CTP to target miRNA106a to HCMs in the heart will be performed to identify cardiac-specific delivery and efficacy of miRNA106a.

There are a number of CamK2 variants.

There are 4 genes α, β, γ, δ. We did not analyze the alpha or beta types because they have not been significantly implicated in the literature to cause HF. On the contrary, the δ variant is widely accepted as causing hypertrophy in cardiomyocytes when over-expressed. The δ variant has two splice forms; δ**_B_** and δ**_C_**. Both are implicated in heart failure [24]. We did not delineate between these two because they would share the same 3′UTR. Thus, miR106a would target both.

Based on the present data, we show that the hypertrophic response to PE and Ang2 can be reversed with CTP-miRNA106a and predict that CTP-miRNAs17, 20a, and 93 should respond similarly by targeting mRNAs that are known to cause hypertrophy when misregulated (i.e., CaMKIIδ and HDAC4). CTP can deliver miRNAs efficiently and specifically to HCMs. Further work using miRNAs 17, 20a and 93 using CTP as a cardiomyocyte-specific vector and cardiac physiology experiments need to be carried out.

## Figures and Tables

**Figure 1 pharmaceuticals-15-00871-f001:**
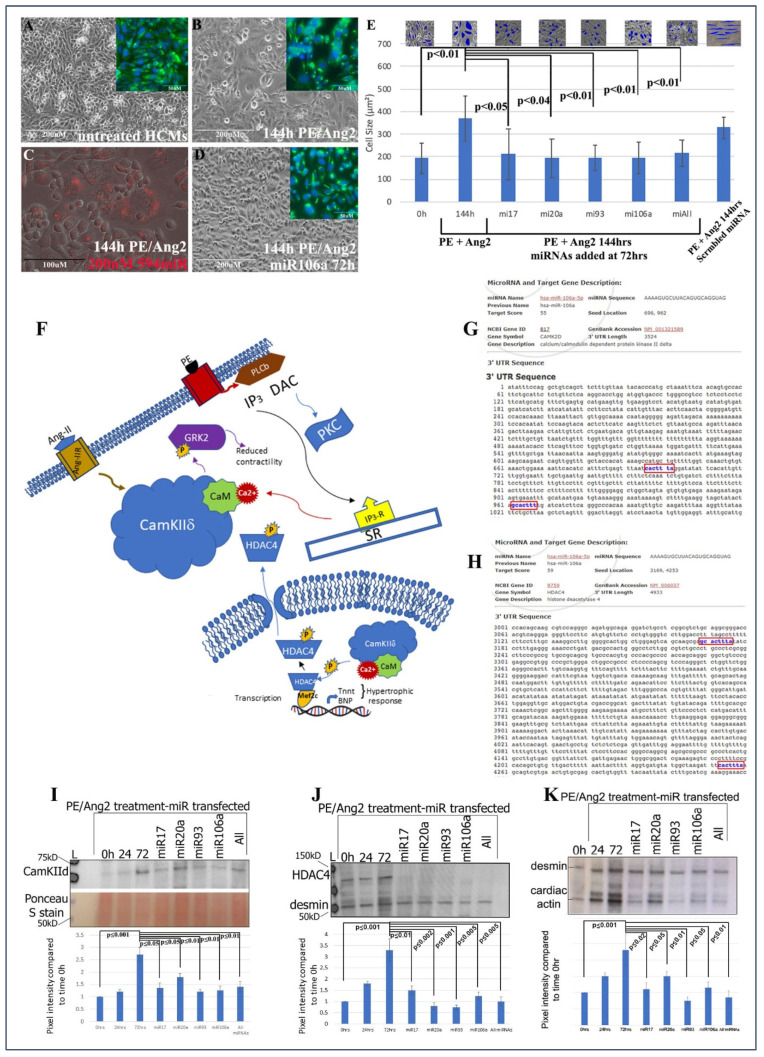
Treating cardiomyocytes with PE and Ang II causes hypertrophy, while treating hypertrophic cells with miRNAs reverts hypertrophic HCMs back to normal size. (**A**) Microscopic images of untreated HCMs. (**B**) HCMs treated with PE/AngII and (**D**) HCMs treated with PE/AngII, followed by miRNA106a. Insets in (**A**,**B**,**D**) show anti-desmin (cardiac) staining in same cells. (**C**) Control using a fluorescently tagged miRNA shows transfection efficiency is ~90%. The size of the cells was measured to quantify hypertrophy (**E**). Insets show Image J measurements of cell size. All cells were measured. Each blue shape represents how the area of cells was measured. All four RNAs were able to reduce hypertrophic morphology. Control scrambled miRNA did not rescue hypertrophic response to PE/Ang2. (**F**) Diagram illustrating the role of CamKIIδ in hypertrophy devised from [17,25,26,27,28]. (**G**,**H**) In silico predicted targeting sites (blue font) for CamKIId and HDAc4. (**I**–**K**) Western blot analyses of CamKIIδ (**I**), HDAc4 (**J**), and a downstream gene of CamKIIδ, cardiac actin (**K**-desmin is used as a loading control). In each case, PE/Ang2 treatment results in increased expression over time. Transfection of each miRNA results in suppression of protein expression within 72 h even in the presents of PE/Ang2. *n* = 6 for each experiment. Note: Image J analyses of Ponceau S-stained blots were used as the loading control for all Westerns. Ponceau S staining is shown in K because desmin and CaMK2d are similar in molecular weights. The lower portion of the blot in (**I**) shows an example of a Ponceau S control. Western Blot repeats can be found in Appendix A.

**Figure 2 pharmaceuticals-15-00871-f002:**
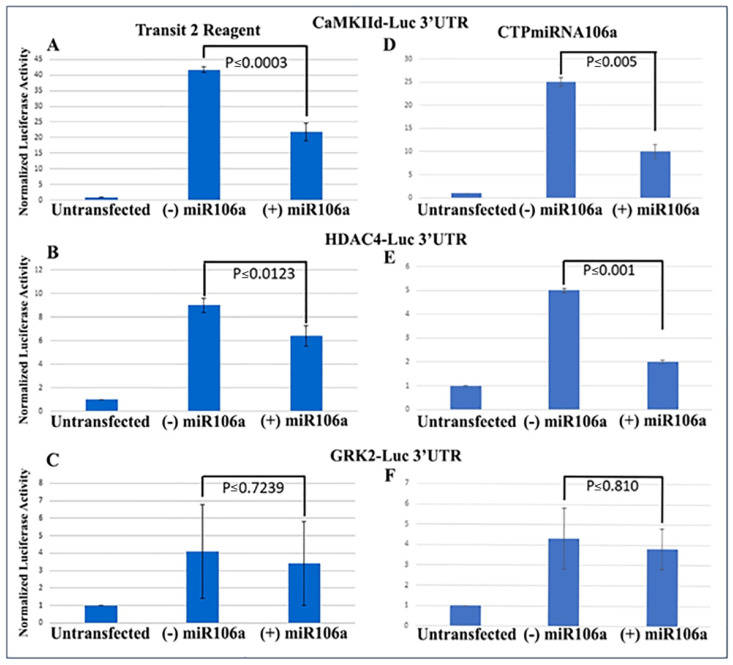
3′UTR-luciferase assays confirm miRNA targets. (**A**–**C**) CamKIIδ (**A**), HDAc4 (**B**) and GRK2 (**C**) 3′UTR-luciferase constructs were transfected using Mirus Transit2 reagent, followed 24 h later transfection using Mirus Transit2 of miRNA106a. Luc activity was significantly decreased for both CamKIIδ and HDAc4 but not GRK2 (GRK2 is not a target of miRNA106a). (**D**–**F**) Similar to (**A**–**C**), each 3′UTR construct was transfected using Mirus Transit2; however, the CTPmiRNA106a was added 24 h later *without* a transfection reagent. Significant reductions in CamKIIδ and HDAc4 were observed but not GRK2. *n* = 6 for each experiment.

**Figure 3 pharmaceuticals-15-00871-f003:**
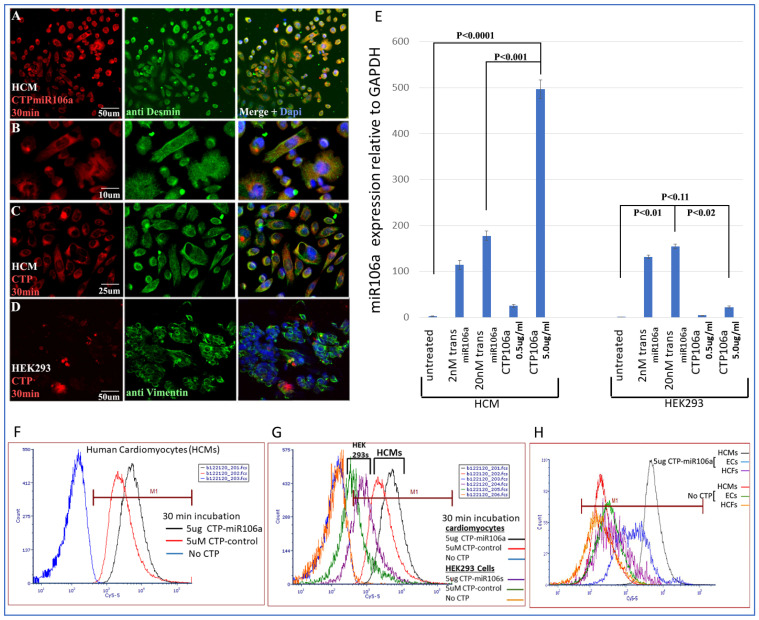
Cy5.5-CTPmiRNA106a was introduced to HCMs, HEK293 cells, Cardiac fibroblasts, and endothelial cells. (**A**–**D**) Fluorescence analyses of CTPmiR106a and CTP-Fluor (RED) reveal its entry into HCMs (**A**–**C**) but not HEK293 cells (**D**). Green stain represents controls for HCM (desmin) and HEK293 cells (vimentin). Dapi (blue) staining identifies nuclei. (**E**) Real-time RT-PCR identifies miRNA106a delivered to HCMs in a dose-dependent manner. A total of 5.0 ug/mL of CTPmiR106a resulted in >5× more expression than 0.5 ug/mL CTPmiR106a. A total of 5 ug/mL CTP106a also showed significantly more expression of miRmiR106a than 2 nM or 200 nM miRNA106a transfected using lipofectamine 3000. Note: HEK293 cells did not take up CTPmiR106a. (**F**,**G**) FACS analysis of HCMs treated with CTPmiR106a (black line) or CTP alone (Red Line). Blue curve is untransfected HCMs shifted left showing no fluorescent activation. (**G**) Same FACs analyses as in (**F**) except HEK293 data are added. Orange and green curves show very little fluorescent activation compared to HCMs. (**H**) CTPmiRNA106a was added to media containing endothelial cells (Blue line) or cardiac fibroblasts (purple line) resulting in no overlap with cardiac fibroblasts and only a slight overlap of signal with endothelial cells. Control No CTP cells show virtually no signal. N = 9 for real-time PCR and N = 3 for FACs and IF experiments.

**Figure 4 pharmaceuticals-15-00871-f004:**
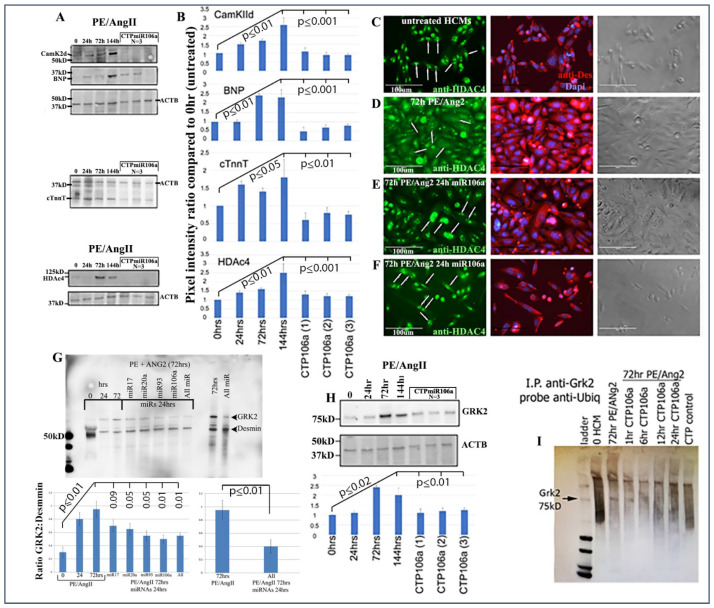
CTPmiRNA106a reverses PE/Ang2-induced hypertrophic responses. (**A**) PE/Ang2 induces expression of CamKIIδ, HDAc4, BNP, and cTNNT, while CTPmiRNA106a suppresses and returns an expression of these proteins to normal (untreated) levels. (**B**) Graphs are culminated from at least six experiments. Significance (at least *p* < 0.05) is observed between each CTPmiRNA106a-treated set and 144 h PE/Ang2-treated HCMs. No significance is observed between each CTPmiRNA106a and untreated HCMs. (**C**–**F**) HDAc4 is nuclear in untreated HCMs (**C**-arrows). (**D**) PE/Ang2 induces CamKIIδ resulting in HDAc4 translocation to the cytoplasm (arrows). (**E**,**F**) Transfection of miRNA106a (**E**) or culture in CTPmiRNA106a (**F**) reverses translocation of HDAc4 back to the nucleus (arrows). Note: compared to untreated cells (**C**), PE/Ang2 causes a marked increase in cell size (**D**), which is reversed after miRNA treatment (**E**,**F**). (**G**) GRK2 increases expression in response to PE/Ang2. Transfection using *Trans*IT-X2 of each miRNA or in combination reverses GRK2 expression back to untreated levels. (**H**) CTPmiRNA106a also reverses increased GRK2 expression caused by PE/Ang2. (**I**) Similar to [30], GRK2 antibody (Mouse) was used to immunoprecipitate GRK2 in HCMs. After blotting, anti-ubiquitin (rabbit) was used to identify ubiquitinated GRK2, which results in a smear of polyubiquitinated GRK2 (lane 2). Increase in CamKIIδ after PE/Ang2 treatment results in less ubiquitination of GRK2 (Lane3), which increases as a result of CTPmiRNA106a suppression of CamKIIδ over time. The CTP control lane is CTP without miRNA106a. All lanes were loaded with 500 ng of lysate measured using a Nanodrop machine. The experiment was done 3 times. Western Blot repeats can be found in Appendix A.

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
