# Peer review of "Reversing Cardiac Hypertrophy at the Source Using a Cardiac Targeting Peptide Linked to miRNA106a: Targeting Genes That Cause Cardiac Hypertrophy"

_pharmaceuticals, 2022, doi:10.3390/ph15070871_

Round 1

Reviewer 1 Report

Authors investigated miRNA106a targets genes which have been shown to cause hypertrophy and the method of specific delivery to cardiomyocytes. This method is important to improve the cardiomyocyte specific delivery. However, the reviewer has some concerns. Please consider and answer as below.

  1. Please show methods for qRT-PCR including primers and immunohistochemistry analyses including device (microscope) and software in materials and methods.
  2. Please add references at the top part of 1st paragraph in introduction.
  3. Please show how authors got endothelial cells at cell type in materials and methods.
  4. In materials and methods, there were some mistakes as below.
    1. Abbreviation was incorrect about Diethyl Ether at CTPmiRNA106A. Et2O is correct.
    2. The full description of TEAbc was made twice at Conjugation of miRNA106a to CTP.
    3. Please declare the name of device and software of FACS and catalog number of antibodies at FACs and immunofluorescence.
  5. It is better to explain with the sentence instead of the reference number.
  6. Please show full descriptions of VASP.
  7. The sentences of “STAT3 was a direct target of miRNAs 17, 20a, 93, and 106a when analyzed using Western blots and 3’UTR luciferase assays [8, 19, 39]” and “Previously we showed the importance of miRNAs 17, 20a, 93, and 106a in cardiac development [8, 9, 19, 39, 40]. Most recently, we used CRISPR/CAS9 gene editing technology to knock-out each of the 4 miRNAs in embryonic stem cells [19]. Upon differentiation, we noticed that each miRNA knock-out resulted in changes in expression of genes involved in heart development. Interestingly, a subset of those genes was also found to be associated with HF. For example, upon knocking out miRNA106a, we noticed significant changes in gene expression of brain natriuretic peptide (BNP), ventricular myosin heavy chain, and the gap junction protein Connexin 43 (CXN43). Importantly, when miRNA106a was added back to mir106a-/- ESCs, expression of each gene returned to normal and cardiac differentiation was rescued.” were just self-citation. There was no strong association with this manuscript. Please remove these sentences.

Author Response

Comments by Reviewer #1

Authors investigated miRNA106a targets genes which have been shown to cause hypertrophy and the method of specific delivery to cardiomyocytes. This method is important to improve the cardiomyocyte specific delivery. However, the reviewer has some concerns. Please consider and answer as below.

1. Please show methods for qRT-PCR including primers and immunohistochemistry analyses including device (microscope) and software in materials and methods.

Our Actions: All information has been added to Materials and methods. In blue font

2. Please add references at the top part of 1st paragraph in introduction.

Our Actions: We have added a comprehensive reference for the information in paragraph one.

3. Please show how authors got endothelial cells at cell type in materials and methods.

Our Actions: They are originally obtained from the ATCC by our Cell and Tissue Core facility at the Lombardi Comprehensive Cancer Center (LCCC). We then purchased a vial from the LCCC. I have put the information in the materials and methods in blue font.

 4. In materials and methods, there were some mistakes as below.

    1. Abbreviation was incorrect about Diethyl Ether at CTPmiRNA106A. Et2O is correct.

Our Actions: This has been changed

              2. The full description of TEAbc was made twice at Conjugation of miRNA106a to CTP.

Our Actions: We removed the second description.

              3. Please declare the name of device and software of FACS and catalog number of antibodies at FACs and immunofluorescence.

Our Actions: I have put the name of the FACs devise in the materials and methods. The antibody catalogue numbers have been added.

               4. Please show full descriptions of VASP.

Our Actions: We have added the full description of VASP in blue font.

Reviewer 2 Report

In this manuscript, Gallicano et al. generated a tool by linking the cardiac targeting peptides with miRNAs to treat heart failure. Although this technique is novel and easily tested in cell culture systems efficiently reduced the cardiac hypertrophy shown by cell size and at the molecular level (reduced the hypertrophy markers; CamKII & HDAC), usage of this technique and its effectiveness (CTP linked miRNA delivery) in preclinical and clinical remains unclear/questionable as both are multi-cellular/organ system.

The following questions were raised while reading this manuscript and require responses.

1.    As mentioned in this manuscript, the four miRNAs involved in regulating cardiac differentiation, whether these miRNAs expressions are changed or not during hypertrophy/ heart failure is still unclear. Authors can include these miRNA expressions during human HF or cell culture model HFs in this manuscript.

2.    As desmin used as control for other immunoblots, why is nt it included for CamKII? Moreover, there are two types of CaMKII (α/β), which one was represented in this manuscript?  Which form of CamKII is targeted by miR-106a?

3.    Although authors used four miRNA in Figure1, what is the rationale to choose miRNA-106a for the remaining experiments?

4.    Please include the n=3 uncropped immunoblots for all results in the supplemental section.  

5.    Are there any titration experiments performed to select the 200nM concentration of miRNA? What is the level of expression (fold) of these miRNAs 24hrs after transfection or CTPlinked treatment?

6.    Method section and figure legends are not clear for luciferase experiments, so authors can re-check for all other experiments also.

7.    Include the expansion for PE in the method section.  

Author Response

Comments by Reviewer #2

In this manuscript, Gallicano et al. generated a tool by linking the cardiac targeting peptides with miRNAs to treat heart failure. Although this technique is novel and easily tested in cell culture systems efficiently reduced the cardiac hypertrophy shown by cell size and at the molecular level (reduced the hypertrophy markers; CamKII & HDAC), usage of this technique and its effectiveness (CTP linked miRNA delivery) in preclinical and clinical remains unclear/questionable as both are multi-cellular/organ system.

The following questions were raised while reading this manuscript and require responses. 

  1. As mentioned in this manuscript, the four miRNAs involved in regulating cardiac differentiation, whether these miRNAs expressions are changed or not during hypertrophy/ heart failure is still unclear. Authors can include these miRNA expressions during human HF or cell culture model HFs in this manuscript.

Our Actions: We cite Guan et al. [29] who showed very subtle changes in miRNA17, 20a, and 93 expression changes during induced HF. They do show an increase in miR106a; however, they did not analyze if this increase affected targets such as CaMK2d or HDAc4, which have been shown repeatedly in the literature to cause hypertrophy and eventual HF if their increased expression levels are left unchecked. We do not believe that the subtle increase in expression of miR106a during ANG2/PE-induced hypertrophy that Guan et al [29] observed was high enough to have an effect on CamK2d and HDAC4.

  1. As desmin used as control for other immunoblots, why is it not included for CamKII? Moreover, there are two types of CaMKII(α/β), which one was represented in this manuscript?  Which form of CamKII is targeted by miR-106a?

Our Actions: For the first question, we used the Ponceau S staining here as a loading control because desmin and CamK2d are virtually the same molecular weight. The NIH has begun to accept Ponceau S staining as a sufficient loading control alternative. I have added this information to the figure legend.  Hope this loading control is acceptable. 

We understand that there are a number of CamK2 variants. There are 4 genes a, b, g, d. We did not analyze the alpha, beta or gamma types because they haven’t been significantly implicated in the literature to cause HF. On the contrary, the d variant is widely accepted as causing hypertrophy in cardiomyocytes when over-expressed.  The d variant has two splice forms a dB and dC. Both are implicated in heart failure. I have added a reference for this statement [24]. We did not delineate between these two because they would share the same 3’UTR. Thus, miR106a would target both. I have added this information to the discussion in blue font.

  1. Although authors used four miRNA in Figure1, what is the rationale to choose miRNA-106a for the remaining experiments?

Our Actions: The rationale was based on our previous work showing 1) the dramatic shift in cardiac gene expression after losing miRNA106a via CRISPR/Cas9 in developing cardiomyocytes and 2) the ability of miRNA106a to rescue all cardiac genes in the miRNA Kos. Future work will include the other miRNAs to identify their efficacy in reducing hypertrophy.

  1. Please include the n=3 uncropped immunoblots for all results in the supplemental section.  

Our Actions: We have added the uncropped immunoblots for Westerns in the supplemental figures.

  1. Are there any titration experiments performed to select the 200nM concentration of miRNA? What is the level of expression (fold) of these miRNAs 24hrs after transfection or CTPlinked treatment?

Our Actions: We chose 200nM based on numerous previous publications from my laboratory. Refs [e.g., 8, 39] and one we recently published looking at the ability of miRNAs to down-regulate the spike mRNA in Sars CoV-2 all showed tritrations of 200nM miRNAs as the best concentration for knocking down expression. We show the level of expression of miR106a using CTP and traditional transfection reagent in figure 3 after 24hrs of incubation. The delivery of miRNAs is quick using CTP. The fluorescent images were cells after 1hr of incubation in fluorescent CTPmiR106a or Naked Fluorescent CTP. I have put those citations into the results section in blue font.

  1. Method section and figure legends are not clear for luciferase experiments, so authors can re-check for all other experiments also.

Our Actions: I have rewritten most of the materials and methods for the luciferase assay. As for the figure, I could separate out the CTPmiRNA106a graphs Figure 2D-F from the transfection experiments Figure 2A-C, but I am hoping the reviewer will allow us to keep the figure as is. It shows the effect of the CTP-delivered miRNA side by side with the transfected (traditional transfection) of the miRNA. I have rewritten the figure legend to help explain this in more detail. It is in blue font.

  1. Include the expansion for PE in the method section. 

Our Actions: I think the reviewer meant ‘explanation’ for PE? My colleagues and I are assuming it’s explanation not expansion. I have added a brief explanation for PE in the methods section in blue font.

Reviewer 3 Report

Gallicano and colleauges describe the approach to use a cardiac targeting peptide linked to miRNA106a to reverse cardiac hypertrophy.

The manuscript is overall well written.

Nevertheless, the authors should provide a list of potential miRNA106a targets as they work with isolated cells in culture. Although the approach to use the cardiac targeting peptide seems to work well under their conditions, critical discussion of potential other targets and the potential in vivo relevance of the approach are required: e.g., would the immune system recognize the peptide-miRNA, might other cell types be susceptible – endothelium/smooth muscle/skeletal muscle in vivo?

Unfortunately, the template was not used and the lines in the ms are not numbered. Thus, it will be more difficult to identify the minor comments.

Minor:

End of introduction: “In collaboration with the Zahid Laboratory…” is not appropriate in case of a co-author.

Figure 1: Control scrambled miRNA did not rescue (data not shown because of space). Please re-arrange the Figure to show the data.

Reviewer 4 Report

1.    Correct the following line: (e.g., mircoRNAs) in the abstract and introduction sections

2.    Delete the space between CaMKIII and HDAC4 on page 3 (Three genes/proteins, in particular, CaMKII   , HDAC4)

3.    Describe the full name of PE  and HCMs cells (page 4) in the Methods section or the first time that the word appears in the text

4.    Please incorporate the details of Mirus Transit-2 transfection reagent (Cat and supplier)

5.    Correct the word polyacrylimide in Western blot section

6.    Figure 1 should be redesigned. There is too much information in this image tan could be better described in another section or images of the manuscript.

7.    These lines on page 6 “Mitofusin-2, a mitochondrial membrane, structural maintenance protein was analyzed using Mnf2 monoclonal anyibody from Santa Cruz, Inc.”  should be placed in Western blot section.

8.    Please correct “anitbodies were were isolated” on page 8.

9.    Indicates the statistic analysis performed in the work.

10. The labels in Western blot bands (figure 4A) have a bad resolution

11. Check the grammar and redaction on the document.

12. Please be careful with metric units (for instance mg, mL, etc)

13. The manuscript needs an extensive style edition

Author Response

Comments by Reviewer #4 

  1. Correct the following line: (e.g., mircoRNAs) in the abstract and introduction sections

Our Actions: These have been corrected

  1. Delete the space between CaMKIII and HDAC4 on page 3 (Three genes/proteins, in particular, CaMKII   , HDAC4)

Our Actions: We have checked spacing and corrected the spacing.

  1. Describe the full name of PE  and HCMs cells (page 4) in the Methods section or the first time that the word appears in the text

Our Actions: These have been corrected.

  1. Please incorporate the details of Mirus Transit-2 transfection reagent (Cat and supplier)

Our Actions: The information has been added

  1. Correct the word polyacrylimide in Western blot section

Our Actions: This has been corrected

  1. Figure 1 should be redesigned. There is too much information in this image tan could be better described in another section or images of the manuscript.

Our Actions: I would like to push back on this one comment. While I understand the reviewer’s point that there is a lot of information in figure 1, as is, it provides the rationale and introduces the narrative of the story we are presenting. I hope the reviewer will reconsider and allow the figure to stay as is.

  1. These lines on page 6 “Mitofusin-2, a mitochondrial membrane, structural maintenance protein was analyzed using Mnf2 monoclonal anyibody from Santa Cruz, Inc.”  should be placed in Western blot section.

Our Actions: It has been placed in the Western Blot section.

  1. Please correct “anitbodies were were isolated” on page 8.

Our Actions: This has been corrected

  1. Indicates the statistic analysis performed in the work.

Our Actions: We used paired T-tests. I have put this in the methods in blue font

  1. The labels in Western blot bands (figure 4A) have a bad resolution

Our Actions: We have replaced all illegible lables.

  1. Check the grammar and redaction on the document.

Our Actions: We have had all authors re-read and correct any suspect grammar

  1. Please be careful with metric units (for instance mg, mL, etc)

Our Actions: We have double checked all metric units

Round 2

Reviewer 2 Report

The authors addressed most of this reviewer's comments. However, CaMK2d and all other western blot data were not consistent in all samples between groups, and the authors uploaded n=2 images/ target. The authors of this manuscript used ActB as a control for one membrane and desmin for another membrane/gel of the same target which is confusing. So authors are required to upload n=3 images for each target with appropriate controls. 

There are no n=3 Ponceau staining images in the supplement for Camk2D. Moreover, If ActB was used an alternative control in the raw images then why didnt author use ActB as control for CamK2d in Figure 1? 

Author Response

Comments round 2 by reviewer #2

The authors addressed most of this reviewer's comments. However, CaMK2d and all other western blot data were not consistent in all samples between groups, and the authors uploaded n=2 images/ target. The authors of this manuscript used ActB as a control for one membrane and desmin for another membrane/gel of the same target which is confusing. So authors are required to upload n=3 images for each target with appropriate controls. 

Our Action: I respect what the reviewer is asking me to do, but over the last couple of years I've had a number of students help me with this project. While we always did the Ponceau S staining of the Westerns, they sometimes took pictures of the Ponceau S stained membranes with their phones to show me later. We have those pictures but in some cases you can't see the lanes because of glare or shadows from the lighting. We always did the housekeeping genes for controls as well as counted cell numbers before adding sample buffer to all cell pellets (I did not say this in the materials and methods but I have added this information to the Wetsern section of the Materials and methods section). My point is we did many controls; however, we don't have all 27 ponceau S stained membranes. I hope this doesn't jeopardize the publication.

Secondly, The reviewer has also questioned why we used ActB sometimes and desmin sometimes as controls. To be honest that was because of antibody availability in the laboratory when the experiments were done and I knew that both are considered housekeeping genes for cardiomyocytes. I saw no harm in this as we have published such controls before. I have at least one good example of Ponceau S stained membranes for all of the gels that I have place onto the supplemental Western figure, but I don't have all three for every membrane. I'm not happy with my students, but they did their best to run good controls. 

Three experiments are represented by the actual figure in the paper (n=1) and then the two we uploaded n=2 and n=3 images in the supplemental figure. We hope that makes sense to the reviewer.

Reviewer 4 Report

Correct the word anyibody in page 3.

Author Response

We have corrected the word anybody to antibody.

Round 3

Reviewer 2 Report

The authors responded to all comments from this reviewer. So, the reviewer recommends accepting the manuscript for publication. 

Author Response

Thank you for this good news. I appreciate all of your comments. They made the paper better.